# Consistency-guided Prompt Learning for Vision-Language Models

**Shuvendu Roy, Ali Etemad**
Queen's University, Canada
{shuvendu.roy, ali.etemad}@queensu.ca

## Abstract

We propose Consistency-guided Prompt learning (CoPrompt), a new fine-tuning method for vision-language models. Our approach improves the generalization of large foundation models when fine-tuned on downstream tasks in a few-shot setting. The basic idea of CoPrompt is to enforce a consistency constraint in the prediction of the trainable and pre-trained models to prevent overfitting on the downstream task. Additionally, we introduce the following two components into our consistency constraint to further boost the performance: enforcing consistency on two perturbed inputs and combining two dominant paradigms of tuning, prompting and adapter. Enforcing consistency on perturbed input serves to further regularize the consistency constraint, thereby improving generalization. Moreover, the integration of adapters and prompts not only enhances performance on downstream tasks but also offers increased tuning flexibility in both input and output spaces. This facilitates more effective adaptation to downstream tasks in a few-shot learning setting. Experiments show that CoPrompt outperforms existing methods on a range of evaluation suites, including base-to-novel generalization, domain generalization, and cross-dataset evaluation. On generalization, CoPrompt improves the state-of-the-art on zero-shot tasks and the overall harmonic mean over 11 datasets. Detailed ablation studies show the effectiveness of each of the components in CoPrompt. We make our code available at https://github.com/ShuvenduRoy/CoPrompt.

## 1 Introduction

Vision-language foundation models (e.g., CLIP (Radford et al., 2021)) that are trained on large-scale datasets of image-text pairs have demonstrated excellent generalization capabilities. However, the sheer size of these models can make it challenging to fine-tune them for downstream tasks, especially for small downstream tasks (e.g., few-shot learning), while preserving their ability to generalize (Zhou et al., 2022a;b). To overcome this challenge, various methods have been proposed to fine-tune these large foundation models by adding and tuning extra parameters (such as prompting (Zhou et al., 2022b) or adapter (Gao et al., 2023)), while keeping the pre-trained weights frozen. Prompt-based methods introduce additional

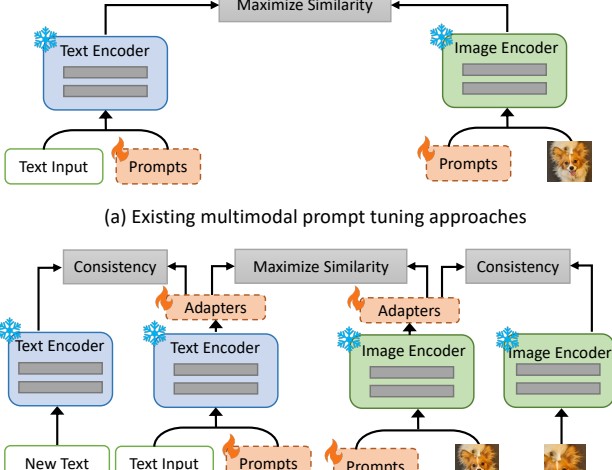

(a) Existing multimodal prompt tuning approaches

(b) Our consistency-guided multimodal prompt tuning (CoPrompt)

Figure 1: Comparison between the CoPompt and existing prompting approach.

tunable weights in the input space (i.e., with image (Bahng et al., 2022), text (Zhou et al., 2022b), or both (Khattak et al., 2023a) embeddings), whereas adapter-based methods (Gao et al., 2023) add learnable weights inside the network, typically near the prediction head.

Despite these advancements in few-shot fine-tuning, it is still a challenge to maintain the generalization capability of the pre-trained model, let alone improve it. In fact, it has been shown that improvements in few-shot performance often result in a drop in zero-shot capabilities (e.g., CoOp (Zhou et al., 2022a)). This is mostly caused by severe overfitting on newly introduced parameters during few-shot fine-tuning, resulting in a significant deviation from the foundation model's original behaviour.

In this work, we propose Consistency-guided Prompt learning (CoPrompt), a new fine-tuning method for vision-language models that reduces the overfitting problem and improves generalization by preventing the trainable model's embeddings from deviating too far from the pre-trained model's embedding when learning a new task. More specifically, we enforce a consistency constraint on both the language and image branches between the trainable and pre-trained models, as illustrated in Figure 1. Unlike consistency regularization in self-supervised learning, where perturbed inputs train two learnable encoders, our approach focuses on maintaining consistency between a learnable encoder and a pre-trained one. This method effectively enables knowledge distillation from the frozen encoders to the learnable ones, thus maintaining the generalization strength of the pre-trained base model while tackling a new task in a few-shot scenario. Furthermore, we introduce two additional components to improve the proposed consistency constraint. First, we enforce consistency on two perturbed inputs instead of the same input to further regularize the consistency constraint, effectively improving generalization. On the 'text' branch, we use a pre-trained large language model (LLM), GPT (Brown et al., 2020), to generate a more detailed and descriptive sentence from an input prompt text of a generic format (a photo of a 'class'). We then enforce consistency between the learnable and pre-trained text encoder on the representations of these two sentences. On the image branch, we apply augmentations on an input image to generate two perturbed images. Secondly, we integrate the two leading paradigms of tuning, namely prompting (Khattak et al., 2023a) and adapters (Gao et al., 2023). This integration offers enhanced tuning flexibility in both the input and output spaces, facilitating more effective learning of new tasks in a few-shot setting. While the fundamental notions of adapters (Gao et al., 2023) and prompts (Khattak et al., 2023a) are explored in the literature, prior works have not been able to successfully combine them for improved performance since models tend to overfit due to the additional learnable parameters, thus losing generalizability (we empirically demonstrate this effect in the ablation study). By integrating both the prompt and adapter methods, as well as applying a consistency constraint, we can optimize additional parameters to enhance performance on new tasks. At the same time, the consistency constraint helps to either maintain or potentially improve the model's capabilities in zero-shot learning scenarios.

Extensive experiments on three common evaluation settings, including base-to-novel generalization, domain generalization, and cross-dataset evaluation, demonstrate the strong performances of Co-Prompt. In the base-to-novel generalization task, CoPrompt outperforms existing methods on 11 benchmark datasets, achieving a 1.13% improvement in novel classes and a 0.51% improvement in the harmonic mean over the previous SOTA. Importantly, our improvements do not come at the expense of the performance of the base class, which also exhibits robust performance. Additionally, CoPrompt achieves considerable improvements over existing methods on cross-dataset evaluation. An extensive ablation study confirms the importance of each component of the proposed method. In summary, this paper makes the following contributions: (**1**) We propose a consistency-enforced fine-tuning method for large foundation models that enables learning a new task from a few samples without losing zero-shot generalizability. (**2**) Our method incorporates the knowledge of a pre-trained LLM with consistency constraints on the text branch and data augmentations on the image branch to improve the generalization further. (**3**) Our method combines the two strong paradigms of tuning foundation models, prompting and adapter, into a single framework to improve performance on new tasks. (**4**) We set a new state-of-the-art for a range of evaluation suites, including base-to-novel generalization and cross-dataset recognition.

## 2 RELATED WORK

Recent developments in vision-language models, such as CLIP (Radford et al., 2021), ALIGN (Jia et al., 2021), LiT (Zhai et al., 2022), FILIP (Yao et al., 2021), and Florence (Yuan et al., 2021), have exhibited impressive performance in various vision tasks. However, the enormous size of these models makes it challenging to fine-tune them without losing their generalization. The two commonly used approaches for using a pre-trained model for a downstream task are (a) full fine-tuning and (b) linear probing. However, neither of these methods performs well for foundation models. Full fine-tuning results in a loss of generalization, while linear probing often leads to poor performance

on downstream tasks (Khattak et al., 2023a). Consequently, many recent studies have focused on adapting large foundation models on downstream tasks without changing the pre-trained weights (Zhou et al., 2022a). Existing works in this direction can be categorized into two main groups: Prompting (Zhou et al., 2022b; Khattak et al., 2023a) and Adapter (Gao et al., 2023).

Prompts are typically instructions in the form of text that guide the downstream task. They can either be manually crafted for a specific task or learned automatically. The latter method is called prompt tuning, which was initially introduced by Lester et al. (2021); Li & Liang (2021); Liu et al. (2021). In this context, CoOp (Zhou et al., 2022a) introduced a set of continuous vectors into the text branch's input, which is optimized with the final loss. However, this approach demonstrated poor performance on unseen classes, indicating poor generalization on zero-shot tasks. CoCoOp (Zhou et al., 2022b) improved CoOp's zero-shot performance by explicitly conditioning on the image inputs. ProGrad (Zhu et al., 2023) only updated the prompts where the gradients aligned with the original prompt's knowledge. Bayesian Prompt Learning (Derakhshani et al., 2023) is a prompt learning method that approaches the task from a Bayesian perspective, formulating it as a variational inference problem. ProDA (Lu et al., 2022) proposed a data-driven approach, that learns the soft prompts from a few downstream samples, discovering the task-related content with less bias than manual design. Prompts have also been utilized for dense prediction tasks (Rao et al., 2022). While the earlier works on prompting added prompts only to the text input, some recent works have also explored prompting on the image inputs (Bahng et al., 2022). Later, MaPLe took a multi-modal approach that used prompting on both image and text inputs. This method explicitly ensured mutual synergy between the text and image prompts to discourage learning from unimodal features. Lastly, PromptSRC (Khattak et al., 2023b) introduced prompting on both image and text inputs, but unlike MaPLe, it trains independent learnable prompts for text and image. Additionally, PromptSRC introduced a self-regulating concept for learning more task-agnostic knowledge, which ensures improved generalization.

Another method for tuning foundation models is Adapters. This approach introduces learnable parameters to one or multiple layers of the pre-trained model to transform features (Gao et al., 2023). Adapters are typically added to the upper layers of the network, which can be seen as a learnable transformation module for the pre-trained model. Adapters have also been studied in vision-only models, including dense prediction tasks (Chen et al., 2022). In our work, we combine both prompting and adapter into a single framework that enhances downstream performance. The additional tunable parameters allow for better adaptation to downstream tasks, while the consistency constraint avoids overfitting and ensures better generalization.

## 3 METHOD

In this section, we present the details of the proposed CoPormpt method. First, we discuss the preliminaries on vision-language models and prompting required for our method, followed by the specifics of the proposed CoPrompt method.

### 3.1 PRELIMINARIES

We adopt CLIP (Radford et al., 2021) as the pre-trained vision-language foundation model in our method. CLIP consists of a transformer-based image encoder, $\theta$, and a transformer-based text encoder, $\phi$. CLIP performs zero-shot prediction by freezing the pre-trained encoders and searching for the highest similarity between the embedding of an input image and the embeddings of hand-crafted text prompts for all class names. CLIP generates the hand-crafted text prompt following the template 'a photo of a [category]'. With $C$ being the number of classes, the text embedding for all class names can be represented as $W = \{w_k\}_k^C$, where $w_k = \phi$('a photo of a [category]$_k$'). For an input image $x$, the image embedding is extracted by the image encoder as $z = \theta(x)$. Finally, CLIP makes a zero-shot prediction as follows:

$$p(y|x) = \frac{\exp(sim(z, w_y)/\tau)}{\sum_{k=1}^{C} \exp(sim(z, w_k)/\tau)}, \tag{1}$$

where, $\tau$ is a temperature parameter, and $sim(.)$ is the cosine similarity.

Even though CLIP shows strong zero-shot performance, it requires further tuning to perform well on new downstream tasks. Additionally, the hand-crafted template approach does not perform well universally across domains. To this end, CoOp (Zhou et al., 2022a) proposed a solution by replacing

the hand-crafted prompt with a set of learnable continuous context vectors that generate task-specific text embeddings. More specifically, CoOp replaced the fixed sentence 'a photo of a [category]' with $m$ learnable prompt vectors ($u_k$) of the same dimension as the word embedding of CLIP. These are concatenated with the word embedding of the class name $c_k$, yielding $t_k = \{u_1, u_2, ....u_m, c_k\}$. In multi-modal promoting methods (Khattak et al., 2023a), an input image is first projected to patch embeddings ($\{p_1, p_2, ...p_d\}$), and then learnable context vectors are concatenated, yielding $i = \{v_1, v_2, ..v_m, p_1, p_2, ...p_d\}$, where $d$ is the number of image patches. Consequently, the modified prediction objective of CLIP with learnable prompt can be expressed as:

$$p(y|x) = \frac{\exp(sim(\theta(i), \phi(t_y))/\tau)}{\sum_{k=1}^{C} \exp(sim(\theta(i), \phi(t_k))/\tau)}. \tag{2}$$

In this work, we build upon the multi-modal prompting concept of MaPLe (Khattak et al., 2023a), which utilizes a coupling function $F$, to condition the image prompt on the corresponding text prompt. A major limitation of the existing fine-tuning methods is that the zero-shot generalizability is severely hampered when new learnable parameters are fine-tuned for a specific downstream task which is caused by overfitting on the downstream dataset. In the following section, we introduce CoPrompt, a novel approach that addresses this problem and enhances the performance for both downstream tasks and zero-shot prediction.

### 3.2 CoPrompt: Consistency-guided Prompt Learning

CoPrompt tackles the issue of reduced generalization due to overfitting on the downstream task, by implementing a consistency constraint that ensures the text and image embeddings produced by the trainable model (tunable prompt parameters in both the image and text branches) are not significantly different from those generated by the pre-trained CLIP. To further impose regularization in the consistency constraint, we utilize perturbations to the input for the trainable and pre-trained models. On the language branch, a pre-trained LLM is used to generate more descriptive sentences from the template text input, while on the image branch, we use augmentations. In addition, CoPrompt includes additional trainable parameters by adding adapters on the image and text branches to enhance performance on new downstream tasks. While the consistency constraint of CoPrompt is conceptually similar to the regulating concept of PromptSRC (Khattak et al., 2023b), CoPrompt distinguishes itself through differences in the criteria for the consistency constraint, types of learnable parameters, and implementation specifics. Specifically, independent prompts are the only training parameters in PromptSRC, while CoPormpt tunes the multi-modal prompts and the adapters together. In the language branch, PromptSRC employs handcrafted prompts, while CoPrompt utilizes an LLM to generate more descriptive prompts. Unlike PormptSRC, CoPrompt utilizes cosine loss as the consistency constraint, capturing the angular similarity between vectors rather than relying solely on their magnitude. An overview of CoPrompt is illustrated in Figure 2. The consistency constraint, input perturbation, and adapters of the proposed CoPrompt are discussed in further detail below.

**Consistency constraint.** We use cosine distance as the consistency constraint between the embeddings of the pre-trained and the learnable encoder. However, other similar criteria, like Euclidean distance, can also be used as a constraint. We empirically observe that cosine distance yields the best performance since it captures the angular similarity between vectors rather than relying solely on their magnitude. This constraint is applied on both image and text branches. Following the notation introduced in the preliminaries, we can denote the consistency constraint as:

$$\mathcal{L}_{cc} = 2 - \frac{w_y \cdot \phi(t_y)}{||w_y|| \, ||\phi(t_y)||} - \frac{z \cdot \theta(i)}{||z|| \, ||\theta(i)||}. \tag{3}$$

Here, $y$ is the class label of the input image.

**Input perturbation.** Given the template text 'a photo of a [category]', we use a pre-trained LLM, GPT($\phi_{GPT}$), to generate a more descriptive sentence as $s_k = \phi_{GPT}$('a photo of a [category]$_k$'). For this, we follow the training setup of KgCoOp (Yao et al., 2023). But unlike KgCoOp, we generate a single sentence on the fly rather than generating a pre-defined number of sentences and averaging their embedding. On the image branch, we use an augmentation module $\delta$ to generate perturbed image $x' = \delta(x)$. Now we enforce the consistency between the embedding of the perturbed input to the pre-trained model and the learnable model as:

$$\mathcal{L}_{cc} = 2 - \frac{\phi(s_y) \cdot \phi(t_y)}{||\phi(s_y)|| \, ||\phi(t_y)||} - \frac{\theta(x') \cdot \theta(i)}{||\theta(x')|| \, ||\theta(i)||}. \tag{4}$$

**Adapters.** We incorporate more trainable parameters for better adaptation to new tasks. However, the literature consistently shows that adding excessive tunable parameters does not improve performance and can harm zero-shot performance (Gao et al., 2023; Khattak et al., 2023a). For instance, Maple (Khattak et al., 2023a) achieved the best performance when adding learnable parameters to only nine out of twelve layers of the CLIP backbone. Further additions resulted in decreased performance. Similarly, in adapter-based approaches

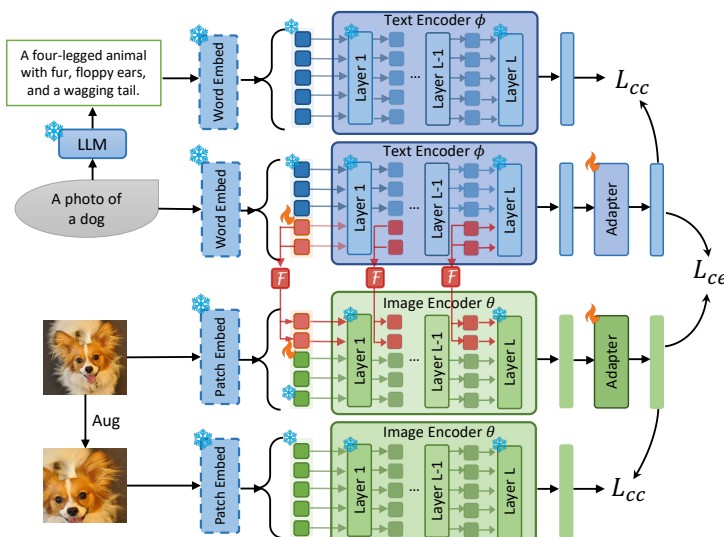

Figure 2: Overview of the proposed CoPrompt.

like CLIP-Adapter (Gao et al., 2023), optimal performance was observed by adding prompts to the text branch alone. Adding prompts to both branches led to overfitting and a subsequent drop in performance.

In this work, we integrate both adapters and prompts to augment the learning capacity. This integration offers enhanced flexibility for tuning in both the input and output spaces. Adapters are trainable parameters that are added on top of the encoder to transform the embedding vector. Following Gao et al. (2023), we define our adapter as two linear layers with non-linearity in between. But unlike Gao et al. (2023), we do not restrict the adapter only to the text branch; rather use it on both. Let $\phi^a$ be the text adapter that takes a text embedding $w_k$ as input and transforms it as $\phi^a(w_k)$. Similarly, $\theta^a$ is the image adapter. Incorporating that into our proposed consistency constraint, the proposed consistency constraint loss can be represented as:

$$\mathcal{L}_{cc} = 2 - \frac{\phi(s_y) \cdot \phi^a(\phi(t_y))}{||\phi(s_y)|| \, ||\phi^a(\phi(t_y))||} - \frac{\theta(x') \cdot \theta^a(\theta(i))}{||\theta(x')|| \, ||\theta^a(\theta(i))||}. \tag{5}$$

**Final loss.** The proposed consistency constraint loss is combined with a supervised loss to form the final loss. We represent the supervised loss as:

$$\mathcal{L}_{ce} = -log \frac{\exp(sim(z, w_y)/\tau)}{\sum_{k=1}^{C} \exp(sim(z, w_k)/\tau)}. \tag{6}$$

Adding both losses with a balancing factor $\lambda$, we get the final loss function of CoPrompt is:

$$\mathcal{L} = \mathcal{L}_{ce} + \lambda \mathcal{L}_{cc} \tag{7}$$

## 4 EXPERIMENTS

### 4.1 EXPERIMENT SETUP

To evaluate the proposed method, we follow the experiment setup and protocols established in CoOp (Zhou et al., 2022a) and subsequent works, such as CoCoOp (Zhou et al., 2022b), and MaPLe (Khattak et al., 2023a). We describe the datasets, training details and evaluation protocol in the Supplementary Section A.1.

### 4.2 BASE TO NOVEL GENERALIZATION

In this section, we present the results of our proposed method on the base-to-novel generalization task. Table 1 presents a detailed comparison of our method with CLIP (Radford et al., 2021), CoOp (Zhou

Table 1: **Comparison with state-of-the-art methods on base-to-novel generalization**. The best accuracies are bolded. HM indicates the harmonic mean.

(a) **Average**

|  | Base | Novel | HM |
|---|---|---|---|
| CLIP | 69.34 | 74.22 | 71.70 |
| CoOp | 82.69 | 63.22 | 71.66 |
| Co-CoOp | 80.47 | 71.69 | 75.83 |
| ProGrad | 82.48 | 70.75 | 76.16 |
| KgCoOp | 80.73 | 73.60 | 77.00 |
| MaPLe | 82.28 | 75.14 | 78.55 |
| PromptSRC | **84.26** | 76.10 | 79.97 |
| CoPrompt | 84.00 | **77.23** | **80.48** |

(b) ImageNet

|  | Base | Novel | HM |
|---|---|---|---|
| CLIP | 72.43 | 68.14 | 70.22 |
| CoOp | 76.47 | 67.88 | 71.92 |
| Co-CoOp | 75.98 | 70.43 | 73.10 |
| ProGrad | 77.02 | 66.66 | 71.46 |
| KgCoOp | 75.83 | 69.96 | 72.78 |
| MaPLe | 76.66 | 70.54 | 73.47 |
| PromptSRC | 77.60 | 70.73 | 74.01 |
| CoPrompt | **77.67** | **71.27** | **74.33** |

(c) Caltech101

|  | Base | Novel | HM |
|---|---|---|---|
| CLIP | 96.84 | 94.00 | 95.40 |
| CoOp | 98.00 | 89.81 | 93.73 |
| Co-CoOp | 97.96 | 93.81 | 95.84 |
| ProGrad | 98.02 | 93.89 | 95.91 |
| KgCoOp | 97.72 | 94.39 | 96.03 |
| MaPLe | 97.74 | 94.36 | 96.02 |
| PromptSRC | 98.10 | 94.03 | 96.02 |
| CoPrompt | **98.27** | **94.90** | **96.55** |

(d) OxfordPets

|  | Base | Novel | HM |
|---|---|---|---|
| CLIP | 91.17 | 97.26 | 94.12 |
| CoOp | 93.67 | 95.29 | 94.47 |
| Co-CoOp | 95.20 | 97.69 | 96.43 |
| ProGrad | 95.07 | 97.63 | 96.33 |
| KgCoOp | 94.65 | 97.76 | 96.18 |
| MaPLe | 95.43 | 97.76 | 96.58 |
| PromptSRC | 95.33 | 97.30 | 96.30 |
| CoPrompt | **95.67** | **98.10** | **96.87** |

(e) StanfordCars

|  | Base | Novel | HM |
|---|---|---|---|
| CLIP | 63.37 | 74.89 | 68.65 |
| CoOp | 78.12 | 60.40 | 68.13 |
| Co-CoOp | 70.49 | 73.59 | 72.01 |
| ProGrad | 77.68 | 68.63 | 72.88 |
| KgCoOp | 71.76 | **75.04** | 73.36 |
| MaPLe | 72.94 | 74.00 | 73.47 |
| PromptSRC | **78.27** | 74.97 | **76.58** |
| CoPrompt | 76.97 | 74.40 | 75.66 |

(f) Flowers102

|  | Base | Novel | HM |
|---|---|---|---|
| CLIP | 72.08 | **77.80** | 74.83 |
| CoOp | 97.60 | 59.67 | 74.06 |
| Co-CoOp | 94.87 | 71.75 | 81.71 |
| ProGrad | 95.54 | 71.87 | 82.03 |
| KgCoOp | 95.00 | 74.73 | 83.65 |
| MaPLe | 95.92 | 72.46 | 82.56 |
| PromptSRC | **98.07** | 76.50 | **85.95** |
| CoPrompt | 97.27 | 76.60 | 85.71 |

(g) Food101

|  | Base | Novel | HM |
|---|---|---|---|
| CLIP | 90.10 | 91.22 | 90.66 |
| CoOp | 88.33 | 82.26 | 85.19 |
| Co-CoOp | 90.70 | 91.29 | 90.99 |
| ProGrad | 90.37 | 89.59 | 89.98 |
| KgCoOp | 90.50 | 91.70 | 91.09 |
| MaPLe | 90.71 | 92.05 | 91.38 |
| PromptSRC | 90.67 | 91.53 | 91.10 |
| CoPrompt | **90.73** | **92.07** | **91.4** |

(h) FGVCAircraft

|  | Base | Novel | HM |
|---|---|---|---|
| CLIP | 27.19 | 36.29 | 31.09 |
| CoOp | 40.44 | 22.30 | 28.75 |
| Co-CoOp | 33.41 | 23.71 | 27.74 |
| ProGrad | 40.54 | 27.57 | 32.82 |
| KgCoOp | 36.21 | 33.55 | 34.83 |
| MaPLe | 37.44 | 35.61 | 36.50 |
| PromptSRC | **42.73** | 37.87 | **40.15** |
| CoPrompt | 40.20 | **39.33** | 39.76 |

(i) SUN397

|  | Base | Novel | HM |
|---|---|---|---|
| CLIP | 69.36 | 75.35 | 72.23 |
| CoOp | 80.60 | 65.89 | 72.51 |
| Co-CoOp | 79.74 | 76.86 | 78.27 |
| ProGrad | 81.26 | 74.17 | 77.55 |
| KgCoOp | 80.29 | 76.53 | 78.36 |
| MaPLe | 80.82 | 78.70 | 79.75 |
| PromptSRC | **82.67** | 78.47 | 80.52 |
| CoPrompt | 82.63 | **80.03** | **81.31** |

(j) DTD

|  | Base | Novel | HM |
|---|---|---|---|
| CLIP | 53.24 | 59.90 | 56.37 |
| CoOp | 79.44 | 41.18 | 54.24 |
| Co-CoOp | 77.01 | 56.00 | 64.85 |
| ProGrad | 77.35 | 52.35 | 62.45 |
| KgCoOp | 77.55 | 54.99 | 64.35 |
| MaPLe | 80.36 | 59.18 | 68.16 |
| PromptSRC | **83.37** | 62.97 | 71.75 |
| CoPrompt | 83.13 | **64.73** | **72.79** |

(k) EuroSAT

|  | Base | Novel | HM |
|---|---|---|---|
| CLIP | 56.48 | 64.05 | 60.03 |
| CoOp | 92.19 | 54.74 | 68.69 |
| Co-CoOp | 87.49 | 60.04 | 71.21 |
| ProGrad | 90.11 | 60.89 | 72.67 |
| KgCoOp | 85.64 | 64.34 | 73.48 |
| MaPLe | 94.07 | 73.23 | 82.35 |
| PromptSRC | 92.90 | 73.90 | 82.32 |
| CoPrompt | **94.60** | **78.57** | **85.84** |

(l) UCF101

|  | Base | Novel | HM |
|---|---|---|---|
| CLIP | 70.53 | 77.50 | 73.85 |
| CoOp | 84.69 | 56.05 | 67.46 |
| Co-CoOp | 82.33 | 73.45 | 77.64 |
| ProGrad | 84.33 | 74.94 | 79.35 |
| KgCoOp | 82.89 | 76.67 | 79.65 |
| MaPLe | 83.00 | 78.66 | 80.77 |
| PromptSRC | **87.10** | 78.80 | 82.74 |
| CoPrompt | 86.90 | **79.57** | **83.07** |

Table 2: Performance of CoPrompt on cross-dataset evaluation and its comparison to existing methods. Here, the model is trained on the ImageNet dataset and evaluated on ten other datasets in a zero-shot setting.

|  | Source | Target | | | | | | | | | | |
|---|---|---|---|---|---|---|---|---|---|---|---|---|
|  | ImNet | Caltech | Pets | Cars | Flowers | Food | Aircraft | SUN397 | DTD | EuroSAT | UCF | *Ave.* |
| CoOp | **71.51** | 93.70 | 89.14 | 64.51 | 68.71 | 85.30 | 18.47 | 64.15 | 41.92 | 46.39 | 66.55 | 63.88 |
| Co-CoOp | 71.02 | 94.43 | 90.14 | 65.32 | 71.88 | 86.06 | 22.94 | 67.36 | 45.73 | 45.37 | 68.21 | 65.74 |
| MaPLe | 70.72 | 93.53 | 90.49 | 65.57 | 72.23 | 86.20 | 24.74 | 67.01 | 46.49 | 48.06 | 68.69 | 66.30 |
| Bayesian Prompt | 70.93 | 93.67 | 90.63 | 65.00 | 70.90 | 86.30 | **24.93** | 67.47 | 46.10 | 45.87 | 68.67 | 65.95 |
| PromptSRC | 71.27 | 93.60 | 90.25 | **65.70** | 70.25 | 86.15 | 23.90 | 67.10 | 46.87 | 45.50 | 68.75 | 65.81 |
| CoPrompt | 70.80 | **94.50** | **90.73** | 65.67 | **72.30** | **86.43** | 24.00 | **67.57** | **47.07** | **51.90** | **69.73** | **67.00** |

et al., 2022a), CoCoOp (Zhou et al., 2022b), ProGrad (Zhu et al., 2023), KgCoOp (Yao et al., 2023), MaPLe (Khattak et al., 2023a), and PromptSRC (Khattak et al., 2023b). We have highlighted the best results in bold. As we see from the average over all datasets (Table 1a), CoPrompt outperforms all existing methods in the harmonic mean of novel and base categories. Our method demonstrates strong zero-shot generalization, with an improvement of 2.09% on novel categories over MaPLe, and 1.13% over PromptSRC. Apart from MaPLe and PromptSRC, no existing method outperformed the pre-trained CLIP (which was not fine-tuned), indicating the difficulty of maintaining zero-shot performance while learning a new task in a few-shot setting. Along with a large improvement in zero-shot performance, CoPormpt also shows strong few-shot performance on base categories. This confirms that improvement in zero-shot performance does not come at the cost of few-shot performance or vice versa. On a harmonic mean, CoPrompt provides a 1.93% improvement over MaPLe and 0.51% over PromptSRC. Observing the HM of individual datasets, we find that CoPrompt outperforms all existing methods on 8 out of 11 datasets.

## 4.3 CROSS-DATASET EVALUATION

In Table 2, we present the results for cross-dataset evaluation. Here, the model is fine-tuned on a source dataset (ImageNet) and evaluated on target datasets in a zero-shot manner. As we see,

CoPrompt shows improvements on 8 out of 10 datasets. Overall, CoPromt provides an average accuracy of 67.0%, which is 0.70% higher than MaPLe, and 1.29% higher than PromptSRC. While PromptSRC shows competitive performance in the base-to-novel generalization task, its performance is considerably lower than CoPrompt in the cross-dataset evaluation.

## 4.4 DOMAIN GENERALIZATION

We present the results for domain generalization in Table 3. Here, the original ImageNet dataset is used as the source dataset to fine-tune the model. The model is then tested on four other variants of ImageNet that come from different distributions. In this evaluation, CoPrompt demonstrates comparable performance to the existing methods, performing only 0.02% and 0.23% lower than Bayesian Prompt and PromptSRC, respectively.

Table 3: Performance on domain generalization.

| | Source | Target | | | | |
|---|---|---|---|---|---|---|
| | ImNet | ImNetV2 | ImNetS | ImNetA | ImNetR | Ave. |
| CLIP | 66.73 | 60.83 | 46.15 | 47.77 | 73.96 | 57.17 |
| UPT | **72.63** | 64.35 | 48.66 | 50.66 | 76.24 | 59.98 |
| CoOp | 71.51 | 64.20 | 47.99 | 49.71 | 75.21 | 59.28 |
| Co-CoOp | 71.02 | 64.07 | 48.75 | 50.63 | 76.18 | 59.90 |
| ProGrad | 72.24 | 64.73 | 47.61 | 49.39 | 74.58 | 59.07 |
| KgCoOp | 71.20 | 64.10 | 48.97 | 50.69 | 76.70 | 60.11 |
| MaPLe | 70.72 | 64.07 | 49.15 | 50.90 | 76.98 | 60.26 |
| Bayesian Prompt | 70.93 | 64.23 | 49.20 | **51.33** | 77.00 | 60.44 |
| PromptSRC | 71.27 | **64.35** | **49.55** | 50.90 | **77.80** | **60.65** |
| CoPrompt | 70.80 | 64.25 | 49.43 | 50.50 | 77.51 | 60.42 |

## 4.5 ABLATION STUDY

**Main ablation.** In this section, we present an ablation study by removing different components of the proposed method to understand the importance of each of them. We show the results of these experiments in Table 4, where 'Cons.', 'In. Pert.', and 'Adp.' represent the consistency constraint, input perturbations, and adapter, respectively. For reference, in the first row of the table, we present the final performance of CoPrompt, which has a harmonic mean of 80.48%. In the first ablation experiment, we eliminate the adapters from CoPrompt, leading to an accuracy of 80.02% (a performance drop of 0.46%). This highlights the importance of adapters in CoPrompt. Next, we remove the input perturbations, effectively enforcing consistency between the trainable and pre-trained encoder for the same image and text input. This results in an accuracy of 79.56%, which is a 0.92% drop in performance, suggesting the high importance

Table 4: Ablation Study

| Cons. | In. Pert. | Adp. | Base | Novel | HM |
|---|---|---|---|---|---|
| ✓ | ✓ | ✓ | 84.00 | 77.23 | 80.48 |
| ✓ | ✓ | ✗ | 83.40 | 76.90 | 80.02 |
| ✓ | ✗ | ✓ | 83.01 | 76.39 | 79.56 |
| ✓ | ✗ | ✗ | 82.90 | 76.36 | 79.50 |
| ✗ | ✗ | ✓ | 83.10 | 74.31 | 78.45 |
| ✗ | ✗ | ✗ | 82.28 | 75.14 | 78.55 |

of input perturbations in CoPrompt. Finally, while we are interested in understanding the importance of the consistency constraint, we can not only remove this component since the input perturbation is also a part of it. So, to understand the impact of this component, we perform two separate studies. First, we remove both the input perturbations and the adapters, which results in an average accuracy of 79.50%. This shows that utilizing the consistency constraint alone provides a 0.95% improvement over removing all three components (as shown in the last row of the table). In the second study, we remove the consistency constraint along with the input perturbations, effectively training the adapters and prompts without enforcing consistency. Surprisingly, this results in an accuracy of 78.45%, even lower than when all three components are removed. This is caused by the fact that the adapters introduce new parameters to train, which causes the trainable model to overfit to the few training samples when the consistency constraint is not enforced. This also results in the lowest zero-shot accuracy (74.31%). These two experiments clearly indicate the significance of the consistency constraint in CoPrompt. In the following section, we present an in-depth analysis of each of these components and their performance for different plausible alternates.

**Analysis of consistency constraints.** Here, we present an analysis of the key aspects of the consistency constraint in CoPrompt. While the consistency constraint is applied to both image and text modalities, we focus on understanding the impact of the consistency constraint on image and text modalities individually. The results of applying consistency on individual modalities are presented in Table 5a. The findings reveal that enforcing consistency on text representations has a greater significance compared to that on image representations. Specifically, using a text-only constraint results in a 0.46% drop in performance, while an image-only constraint leads to a 0.89% decrease. The highest performance is achieved when consistency is enforced on both modalities. Next, we

Table 5: Analysis of different components of CoPrompt.

(a) Cons. modalities.

| Consistency | Accuracy |
|---|---|
| Image only | 79.59 |
| Text only | 80.02 |
| Both | 80.48 |

(b) Consistency criterion.

| Criterion | Accuracy |
|---|---|
| Cosine | 80.48 |
| L1 | 80.40 |
| MSE | 79.33 |

(c) Text input.

| Input | Accuracy |
|---|---|
| Same Text | 80.09 |
| LLM (GPT-2) | 80.46 |
| LLM (GPT-3) | 80.48 |

(d) Image input.

| Input | Accuracy |
|---|---|
| Same Image | 80.16 |
| Simple Aug. | 80.48 |
| Hard Aug. | 79.90 |

(e) Adapter choices.

| Adapter | Accuracy |
|---|---|
| Text only | 80.35 |
| Image only | 80.10 |
| Both | 80.48 |

(f) No. of Adapter layers.

| Layers | Accuracy |
|---|---|
| Single layer | 80.40 |
| 2 layers | 80.48 |
| 3 layers | 79.75 |

investigate the influence of different consistency criteria. Table 5b compares the performance when using cosine distance, MSE, and L1 as consistency constraints. The results demonstrate that cosine distance performs the best as a consistency loss, while L1 shows a very similar performance. However, employing MSE leads to a drop in performance.

**Analysis of input perturbations.** In this section, we explore the impact of different input perturbations on the performance of the final model. Table 5c presents the results for using the same text as input compared to using more descriptive text generated by the LLMs (GPT-2 or GPT-3) as input. The findings reveal a 0.39% decrease (compared to GPT-3 generated text) in performance when the same input is used for both the learnable and frozen text encoders. This emphasizes the importance of utilizing LLMs to generate more descriptive text on which to enforce consistency. However, the performance for GPT-2 and GPT-3 are relatively similar, suggesting that the choice of LLM does not have a large impact. Although various LLMs are specialized for generating coherent and meaningful sentences on complex topics, our focus is on generating a single sentence to describe a category name. In this specific context, the choice among different LLMs does not result in a substantial difference.

Similarly, we conduct a study on the image inputs, as shown in Table 5d. We compare the results when using the same image, simple augmented image, and hard augmented image as inputs. Consistent with the observations on text analysis, using the same image input shows a decline in performance since it fails to provide a sufficiently discriminative signal for learning. On the contrary, we expect hard augmentations to yield the best discriminative features for learning and, consequently, the highest accuracy. However, the results indicate that simple augmentations (random resized crop, horizontal flip) outperform hard augmentations (RandAug (Cubuk et al., 2020)). We believe using hard augmentations leads to significant deviations in the image embeddings, causing them to diverge from the corresponding text embeddings. Consequently, this results in performance degradation.

**Analysis of adapters.** Lastly, we delve into several important factors concerning the design of the adapters. First, we present the results of incorporating adapters on different modalities, as shown in Table 5e. Consistent with previous findings (Gao et al., 2023), we observe that adding an adapter to the text branch yields more benefits compared to the image branch (80.35% compared to 80.1%). However, contrary to the findings by Gao et al. (2023), we do not observe a drop in performance when adapters are employed on both modalities. In fact, the highest accuracy is achieved when adapters are used on both the image and text branches. This emphasizes that while naive few-shot tuning does not benefit from utilizing adapters on both branches, the proposed consistency-guided tuning approach facilitates learning via more tunable parameters on both modalities.

We also explore the impact of the number of linear layers used in the adapter design. We evaluate three different configurations: a single-layer adapter, a 2-layer adapter as proposed by Gao et al. (2023), and a 3-layers

Table 6: Performance of CoPrompt for different values of $\lambda$.

| | ImNet | Caltech | Pets | Cars | Flowers | Food | Aircraft | SUN397 | DTD | EuroSAT | UCF |
|---|---|---|---|---|---|---|---|---|---|---|---|
| 0.0 | 73.47 | 96.01 | 96.56 | 73.46 | 82.56 | 91.39 | 36.50 | 79.74 | 68.16 | 82.34 | 80.77 |
| 0.01 | 73.71 | 96.11 | 96.64 | 73.66 | 82.72 | 91.60 | 36.79 | 79.83 | 68.27 | 83.48 | 80.91 |
| 0.1 | 73.82 | 96.22 | **96.87** | 74.44 | 84.15 | **91.73** | 37.60 | 79.95 | 67.42 | **85.84** | 81.93 |
| 1.0 | 74.05 | 96.44 | 96.86 | 75.43 | 84.99 | 91.43 | 38.69 | 80.40 | 70.09 | 84.04 | 82.57 |
| 2.0 | 74.14 | 96.41 | 96.77 | 75.28 | 84.89 | 91.40 | **39.76** | 80.72 | 72.25 | 81.46 | **83.07** |
| 8.0 | **74.33** | **96.55** | 96.84 | **75.66** | **85.71** | 91.43 | 39.37 | **81.31** | **72.79** | 78.63 | 82.77 |
| 10.0 | 73.22 | 95.65 | 96.06 | 73.23 | 82.25 | 90.37 | 37.49 | 80.15 | 71.17 | 77.25 | 82.11 |

adapter. The results are presented in Table 5f. The findings indicate that the 2-layer adapter slightly outperforms the single-layer design. This suggests that incorporating an additional layer allows for capturing more complex relationships and improving performance to some extent. However, using a 3-layer adapter exhibits a significant drop in performance. We believe this is because adding too many linear layers in the adapters can introduce an excessive number of parameters, which may re-introduce the overfitting problem to the limited training examples available in few-shot settings.

**Sensitivity study.** In this section, we present a sensitivity analysis of the proposed method to some of its key parameters. First, we investigate the impact of the weight factor ($\lambda$) of the consistency constraint loss. From Table 6, we observe that higher values of $\lambda$ lead to better accuracy, indicating the high importance of the consistency constraint. Specifically, we achieve the best accuracy for $\lambda = 8$ on 6 out of 11 datasets and close to the best accuracy on the remaining datasets, except for EuroSAT dataset. Factors such as data distribution and its similarity to the pre-training dataset of CLIP influence the value of $\lambda$, resulting in different optimal values for different datasets. The performance does not improve for $\lambda > 8$.

Next, we explore how the performance of CoPrompt varies with different numbers of prompt layers and present the results in Table 7a. While previous works like MaPLe (Khattak et al., 2023a) have shown the best results with 9 prompt layers, we observe that adding prompts on all 12 layers of the encoder leads to the best accuracy. The consistency constraint introduced in CoPrompt enables us to train more parameters without overfitting. Table 7b shows the results for different numbers of training epochs. We find that the best performance is achieved with 8 epochs.

Table 7: Sensitivity studies.

(a) Acc vs Layers.

| Layers | Acc. |
|---|---|
| 3 | 78.77 |
| 6 | 79.05 |
| 9 | 80.15 |
| 12 | **80.48** |

(b) Acc vs Epochs.

| Epochs | Acc. |
|---|---|
| 3 | 79.54 |
| 5 | 80.24 |
| 8 | **80.48** |
| 10 | 80.02 |

### 4.6 PARAMETERS AND COMPUTATIONAL COMPLEXITY

Following standard practice in the literature, we use the ViT-B/16 backbone of CLIP, which has 149.62M parameters (Table 8). MaPLe introduced an additional 3.55M learnable parameters, resulting in a total of 153.17M parameters. The prompt module of CoPrompt contains 4.74M parameters, and the adapter module contains only 0.26M parameters. Thus, CoPrompt has a total of 154.62M parameters, only less than 1% parameter increase over previous SOTA MaPLe and a 3.34% parameter increase over CLIP.

Table 8: Comparison of total and learnable parameters for different methods.

| Model | Total param. | Learnable param. |
|---|---|---|
| CLIP (ViT-B/16) | 149.62M | - |
| MaPLe | 153.17M | 3.55M |
| CoPrompt | 154.62M | 4.74M |

In terms of training, CoPrompt has around 2x FLOPs (required to generate predictions of the pre-trained model) and takes around 25% more training time than MaPLe (on a single Nvidia V100 GPU). For example, one epoch of training on the Flower102 dataset takes 2 minutes and 9 seconds compared to 1 minute and 43 seconds for MaPLe on a single GPU. However, the *inference* time and FLOPs are almost the same as the MaPLe. For a fair comparison with MaPLe, we perform another experiment by training CoPrompt with an identical computational budget to that of MaPLe by reducing the number of training epochs. Under this configuration, CoPrompt achieves an accuracy of 80.01%, which still is a 1.46% improvement over MaPLe.

## 5 CONCLUSION

We present a novel tuning method for large vision-language foundation models that enhances their performance in downstream tasks and also improve zero-shot generalization. CoPrompt is a carefully designed method with three important components that reduce the overfitting problem during fine-tuning. Through extensive evaluations across three different tasks, CoPrompt demonstrated its effectiveness in few-shot learning, zero-shot learning, cross-dataset, and domain generalization tasks, surpassing the existing state-of-the-art by a significant margin. Additionally, the study includes extensive ablation analysis to confirm the effectiveness of each proposed component and explore feasible alternatives. We believe that our consistency-guided tuning approach will have a substantial impact on tuning vision and vision-language models for various applications.

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

## A APPENDIX

### A.1 EXPERIMENT SETUP

In this section, we present further details regarding our experimental setup. First, we discuss the dataset and implementation specifics, followed by a discussion on the evaluation protocol.

#### A.1.1 DATASETS AND TRAINING DETAILS

We evaluate our model's performance on 11 datasets that cover various recognition tasks, including generic object classification datasets such as ImageNet (Deng et al., 2009) and Caltech101 (Fei-Fei et al., 2004), fine-grained recognition datasets such as OxfordPets (Parkhi et al., 2012), Stanford-Cars (Krause et al., 2013), Flowers102 (Nilsback & Zisserman, 2008), Food101 (Bossard et al., 2014), and FGVCAircraft (Maji et al., 2013), as well as scene recognition dataset SUN397 (Xiao et al., 2010), action recognition dataset UCF101 (Soomro et al., 2012), satellite-image classification dataset EuroSAT (Helber et al., 2019), and texture recognition dataset DTD (Cimpoi et al., 2014). Moreover, to evaluate domain generalization, we utilize four variants of the ImageNet dataset, including ImageNetV2 (Recht et al., 2019), ImageNet-Sketch (Wang et al., 2019), ImageNet-A (Hendrycks et al., 2021b), and ImageNet-R (Hendrycks et al., 2021a).

We adopt CLIP (ViT-B/16) (Radford et al., 2021) as the pre-trained vision-language backbone to evaluate our proposed method. We fine-tune the model in few-shot settings with 16 samples per class for all known classes. The model is trained with an SGD optimizer for 8 epochs, using a batch size of 4 and a learning rate of 0.035. We report the average accuracy over 3 runs on the individual experiments. The training is conducted on a single Nvidia V100 GPU, and the experiment takes slightly less than a day to train and evaluate on all 11 datasets.

#### A.1.2 EVALUATION PROTOCOLS

**Base-to-novel class generalization.** The most prevalent approach for evaluating performance in zero-shot scenarios while fine-tuning on a few-shot setting is through base-to-novel generalization testing. This involves partitioning the dataset into known (base) and unseen (novel) classes, followed by training the model using a small number of samples from the base classes. Finally, the model's performance is evaluated on both base (few-shot performance) and novel (zero-shot performance) categories. We also report the harmonic mean (Xian et al., 2017) over known and unknown classes.

**Cross-dataset evaluation.** In this experiment, we assess the zero-shot ability of the model on a cross-dataset evaluation setup. Specifically, we train the model in a few-shot setting on one dataset and subsequently evaluate its performance on ten other unseen datasets that contain unknown categories. We adopt the evaluation protocol from CoCoOp (Zhou et al., 2022b) and MaPLe (Khattak et al., 2023a) for this experiment.

**Domain generalization.** In addition, we assess the model's performance on out-of-distribution generalization. For this experiment, we first fine-tune the model on one dataset and then test the model on different datasets that contain the same classes but from different distributions.

### A.2 ANALYSIS OF THE LEARNED REPRESENTATIONS

In Table 9, we present the calculated distance between the learned representation of different methods (previous SOTA (MaPLe) and proposed CoPrompt) and the original ones from CLIP. This analysis is performed on the ImageNet dataset. For each image, we compute the distance between the normalized output embeddings of CLIP and the tuned model. Additionally, we calculate the distance for the text branch using the template sentence 'a photo of a category_name.' The results from this table clearly show that the representations of CoPrompt remain close to that of CLIP, while MaPLe deviates a lot in both image and text encoder.

Table 9: Distance metrics between the learned representation of different methods and the original ones from CLIP.

| Method | Dist. (for image) | Dist. (for text) |
|--------|-------------------|------------------|
| MaPLe | 0.56 | 0.45 |
| CoPrompt | 0.37 | 0.28 |

## A.3   PERFORMANCE ON OTHER VISION-LANGUAGE MODEL

We perform additional experiments by chang-
ing the backbone of our method CoPrompt,
as well as MaPLe, from CLIP to EVA-CLIP
(Fang et al., 2023). The results are presented
in Table 10, which demonstrate that CoPrompt
also works well with EVA-CLIP as the back-
bone and shows similar improvements over
MaPLe as with CLIP. Overall, the EVA-CLIP
encoder performs slightly better than the CLIP
encoder.

Table 10: Performance on base-to-novel generalization
with EVA-CLIP backbone.

| Method | Backbone | Base Acc | Novel Acc | HM |
|---|---|---|---|---|
| MaPLe | CLIP-B/16 | 82.28 | 75.14 | 78.55 |
| CoPrompt | CLIP-B/16 | 84.00 | 77.23 | 80.48 |
| MaPLe | EVA-CLIP-B/16 | 82.65 | 75.78 | 79.06 |
| CoPrompt | EVA-CLIP-B/16 | 84.29 | 77.50 | 80.75 |

## A.4   TRAINING ADAPTERS AND PROMPT WITHOUT CONSISTENCY CONSTRAINT

We argue that combining adapters and prompt-tuning is
non-trivial due to the overfitting and subsequent loss of
generalization that occurs as a result of the substantial
increase in parameters. To test this, we combine adapters
and prompt-tuning in our framework without the consis-
tency constraint. As observed in Table 11, CoPrompt's
performance of 80.48% drops to 78.45% when prompts
and adapters are trained without the proposed consis-
tency constraint. Similarly, the optimal performance of
MaPLe (Khattak et al., 2023a) is 78.55%, which drops to 77.61% when we increase the trainable
parameters in the form of prompts and adapters.

Table 11: Performance of adapters and prompts
without our consistency constraint.

| Method/ Setup | Accuracy |
|---|---|
| CoPrompt | 80.48 |
| CoPrompt w/o Cons. const. | 78.45 |
| MaPLe | 78.55 |
| MaPLe (12-layer prompt) + Adapter | 77.61 |

## A.5   COMPARISON TO ZERO-SHOT PERFORMANCE OF CLIP

To ensure a fair comparison to the zero-shot performance of CLIP,
we investigate the results for the following two scenarios: (1)
removing the LLM-generated prompts from CoPrompt, and (2)
adding LLM-generated prompts to CLIP for its zero-shot evalu-
ation. The result of the first experiment are presented in Table 5c
(first row), where CoPormpt obtains an accuracy of 80.09%. While
this is a 0.39% drop from the optimal performance of CoPrompt,
it is 8.99% better than the zero-shot performance of CLIP (80.09%
compared to 71.1%). We show the result for the second experiment in Table 12. The result from this
experiment shows that LLM-generated text prompts do not improve the zero-shot classification ability
of CLIP on downstream tasks, but rather drop the performance from 71.70% to 69.14%. This can be
explained by the fact that using a more complex sentence instead of the simple template decreases
the likelihood of the sentence embedding being aligned with the corresponding image, potentially
leading to inaccuracies in CLIP's predictions.

Table 12: Zero-shot evaluation of
CLIP with hand-designed and LLM-
generated prompts.

| Prompt | Accuracy |
|---|---|
| Hand-designed prompt | 71.10 |
| LLM-generated prompt | 69.14 |

## A.6   CONSISTENCY CONSTRAINT ON OTHER METHODS

To evaluate the impact of adding our proposed consistency
approach to other methods, we add it to CoOp (Zhou et al.,
2022a) and present the results in Table 13. We observe that
adding the constancy constraint to CoOp results in a con-
siderable improvement in accuracy for novel classes (from
63.22% to 64.87%), contributing to an overall increase in
the harmonic mean (from 71.66% to 72.75%).

Table 13: Consistency constraint on CoOp.

| Method | Base Acc. | Novel Acc | HM |
|---|---|---|---|
| CoOp | 82.69 | 63.22 | 71.66 |
| CoOp + Cons. | 82.80 | 64.87 | 72.75 |

