# OpenReview forum: "Consistency-guided Prompt Learning for Vision-Language Models"
_ICLR.cc/2024/Conference — ICLR 2024 poster_

### Official Review · Reviewer_8uZQ · 2023-10-30

**Soundness:** 3 good
**Presentation:** 3 good
**Contribution:** 3 good
**Rating:** 6
**Confidence:** 5

**Summary:**

The paper presents an innovative prompt learning technique that integrates a consistency mechanism between trainable and pre-trained models to mitigate the risk of overfitting. This approach employs a consistency term applied to two altered inputs within the text and visual spheres. In text modality, the method leverages existing language models like GPT-2 and GPT-3 to introduce variations, whereas for images, it uses standard image augmentation techniques prevalent in self-supervised learning. The authors have skillfully merged two distinct approaches to adaptation - prompting and adaptation - demonstrating that this synergy, coupled with a consistency loss, enhances the method's ability to generalize. The improved generalization capability of this approach is evident in various prompt learning tasks, including adapting from base to new tasks, cross-dataset evaluation, and domain generalization, with consistent enhancements observed across these applications.

**Strengths:**

The paper is clear and of high quality, with significant numerical results. The authors offer a thorough analysis of their proposed method's various components, which overall appear sensible and well-founded.

**Weaknesses:**

While the paper is clear, and the numerical results are noteworthy, its novelty isn't entirely clear. The paper's self-consistency terms seem similar to those in self-supervised learning (SSL) methods. The authors' claim of differentiating their approach from SSL, where two perturbed inputs within a single encoder are used, doesn't fully convince. In SSL, typically there are two encoders: an online encoder and a momentum encoder. The paper’s pre-trained and trainable encoders appear analogous to SSL’s momentum and online encoders, respectively. The authors should clarify this similarity.

Additionally, the paper omits recent relevant studies like Bayesian Prompt Learning [1] and Prompt Distribution Learning [2], which address overfitting in vision and language models. Discussing these in the related work and comparing them in sections like domain generalization are necessary, especially given that in some cases, such as Bayesian Prompt Learning, they outperform the methods in this paper. For example, in the domain generalization task, the Bayesian Prompt Learning method (%60.44) works better than the paper performance (%60.42).

[1]. Bayesian Prompt Learning for Image-Language Model Generalization, ICCV 2023

[2]. Prompt Distribution Learning, CVPR 2022

**Questions:**

Please see the weaknesses section.

---

> ### Author Response · Authors · 2023-11-17
>
> > The paper's self-consistency terms seem similar to those in self-supervised learning (SSL) methods.
>
> Our model does indeed use the principles of SSL for applying consistency constraint on the vision-language framework. While in general SSL frameworks have two *learnable encoders* which are both trained, in our method, one encoder is *frozen* while the other is learnable. This method effectively enables knowledge distillation from the frozen encoders to the learnable ones, thus maintaining the generalization strength of the pre-trained base model while tackling a new task in a few-shot scenario. In some sense, our approach is somewhat between SSL and distillation. We have included this discussion in the Introduction (Page 2, Paragraph 2).
>
> > Missing recent relevant studies.
>
> Thank you for pointing out two related works which we were not aware of before. We now provide a discussion on these two methods and compare the results as follows:
>
> Bayesian Prompt Learning [1] is a prompt learning method that approaches the task from a Bayesian perspective, formulating it as a variational inference problem. ProDA [2] proposed a data-driven approach, that learns the soft prompts from a few downstream samples, discovering the task-related content with less bias than manual design. Our approach is quite different from these methods since we build over a multi-modal prompt tuning technique and propose three components aiming to improve both few-shot and zero-shot performance.
>
>
> Following, we show the comparison of our method with Bayesian prompt learning. We can't compare the results with ProDA since it adopted a different training protocol (only evaluated on few-shot learning) than most existing methods, including ours.
>
>
>
> | Method | Base Acc. | Novel Acc | HM |
> | ------ | --------- | --------- | --- |
> | Bayesian Prompt | 80.10 | 74.94 | 77.43 |
> | CoPrmpt (ours)  | 84.00 | 77.23 | 80.48 |
>
>
>
>
> | Method          | Caltech | Pets  | Cars  | Flowers | Food  | Aircraft | SUN397 | DTD   | EuroSAT | UCF   | Average |
> |-----------------|---------|-------|-------|---------|-------|----------|--------|-------|---------|-------|---------|
> | Bayesian Prompt | 93.67   | 90.63 | 65.00 | 70.90   | 86.30 | 24.93    | 67.47  | 46.10 | 45.87   | 68.67 | 65.95   |
> | CoPrompt        | 94.50   | 90.73 | 65.67 | 72.30   | 86.43 | 24.00    | 67.57  | 47.07 | 51.90   | 69.73 | 67.00   |
>
>
>
>
> | Method          | ImNetV2 | ImNetS | ImNetA | ImNetR | Average |
> |-----------------|---------|--------|--------|--------|---------|
> | Bayesian Prompt | 64.23   | 49.20  | 51.33  | 77.00  | 60.44   |
> | CoPrompt        | 64.25   | 49.43  | 50.50  | 77.51  | 60.42   |
>
>
>
> As we find from the comparison of base-to-novel generalization, CoPropmt outperforms Bayesian Prompt by 3.05\% in harmonic mean over 11 datasets. It performs better than Bayesian Prompt on both base and novel categories. On the cross-dataset evaluation, CoPrompt outperforms Bayesian Prompt by 1.05\% on average over all datasets. Finally, on domain generalization, CoPrompt performs only 0.02\% lower than Bayesian Prompt on the average over all datasets, but looking into individual datasets, we find that CoPrompt outperforms Bayesian Prompt on all datasets except ImageNetA.
> We now discuss Bayesian Prompt and ProDA in our Related Work section (Page 3, Paragraph 2), and also expand the results in Tables 2 and 3 (Pages 6 and 7) to compare with Bayesian Prompt. We could not include Bayesian Prompt in Table 1 since it did not report the base, novel, and HM results like other methods.

---

> > ### Comment · Reviewer_8uZQ · 2023-11-21
> >
> > I appreciate the authors' detailed explanation of the differences between their proposed method and the SSL method, as well as for comparing it with recent baselines. The results are so convincing and I am satisfied with the responses from the authors and want to maintain my current score.

---

> > > ### Author Response · Authors · 2023-11-21
> > >
> > > Dear Reviewer,
> > >
> > > Thanks so much for your positive feedback regarding our work and results. We’re pleased to note your post-rebuttal recommendation for accepting the paper.

---

### Official Review · Reviewer_sWY6 · 2023-10-31

**Soundness:** 2 fair
**Presentation:** 3 good
**Contribution:** 2 fair
**Rating:** 6
**Confidence:** 4

**Summary:**

This paper presents a consistency-guided prompt learning (CoPrompt) method to transfer CLIP to downstream tasks in a few-shot setting. Experimental results show the capacity of consistency-guided prompt learning to imporve the generalization comparing with the SOTA methods.

**Strengths:**

1. CoPrompt achieves the SOTA results on base-to-novel generalization and cross-domain setting.

2. Ablation studies demonstrate the effectiveness of consistency constrain to prevent overfitting on the downstream tasks.

**Weaknesses:**

1.The prompt learning and adapter learning method mentioned in this paper are introduced by MaPLe and CLIP-Adapter. The primary contribution of this paper only lies in the introduction of consistency constraint learning.  Thus, regarding the method as prompt learning is a bit ambiguous in my opinion.

2. The comparison between CoPrompt and Zero-shot CLIP is not fair enought. The diverse text prompts generated by LLM can imporve the zero-shot classification ability of CLIP on downstream tasks. It is important to consider this aspect when evaluating the performance of  Zero-shot CLIP.

3. On small scale dataset like Eurosat, the higher value of λ leads to worse performance, accroding to Table 7. However, the analysis regarding this observation is missing from the paper. Can CoPrompt reaches better result on Eurosat if λ=0?

**Questions:**

1. What if combined consistency constrain learning with other existing methods, like CoCoOP. Can CoPrompt improve the generalization of CoCoOP?

2. What is the training overhead in terms of time? What is training and test-time inference speed compared with prior methods?

---

> ### Author Response · Authors · 2023-11-17
>
> > Regarding the method as prompt learning is a bit ambiguous in my opinion.
>
> Prompts are the main learnable parameters in our method, and the goal of our solution is to train the learnable prompt while ensuring effective adaptation to the target domain with few labelled samples without losing generalization. This is why we refer to it as a prompt learning method. Kindly note that existing methods in the literature (e.g. KgCoOp and ProGrad) that share similar conceptual frameworks are also considered prompt learning methods.
>
>
> > The comparison between CoPrompt and Zero-shot CLIP is not fair enough. The diverse text prompts generated by LLM can improve the zero-shot classification ability of CLIP on downstream tasks.
>
> To fully investigate the answer to this question, we show the results for the following two scenarios: (1) removing the LLM-generated prompts from CoPrompt and (2) adding LLM-generated prompts to CLIP for its zero-shot evaluation.
>
> The result of the first experiment is presented in Table 5(c) (first row), where CoPormpt obtains an accuracy of 80.09\%. While this is a 0.39\% drop from the optimal performance of CoPrompt, it is 8.99\% better than the zero-shot performance of CLIP (80.09\% compared to 71.1\%). We show the result of the second experiment in the Table below. The result from this experiment shows that LLM-generated text prompts do not improve the zero-shot classification ability of CLIP on downstream tasks, but rather drop the performance from 71.70\% to 69.14\%. This can be explained by the fact that using a more complex sentence instead of the simple template decreases the likelihood of the sentence embedding being aligned with the corresponding image, potentially leading to inaccuracies in CLIP's predictions. We have added this discussion to Appendix A.5 (Page 14).
>
>
> | Prompt | Accuracy |
> | ------ | --------- |
> | Hand-designed prompt | 71.10 |
> | LLM-generated prompt | 69.14 |
>
>
>
> > Can CoPrompt reaches better result on Eurosat if λ=0?
>
> Using λ=0 will effectively turn off the consistency constraint, which, according to the ablation study, leads to sub-optimal performance. In the table below (which is an expansion of Table 6), we show the results for all datasets with λ=0 and another smaller value of λ=0.01, which does not improve the performance of any dataset further. Accordingly, we have expanded Table 6 (Page 8) and added these results.
>
>
>
> | Lambda | ImNet | Caltech | Pets  | Cars  | Flowers | Food  | Aircraft | SUN397 | DTD   | EuroSAT | UCF   |
> |--------|-------|---------|-------|-------|---------|-------|----------|--------|-------|---------|-------|
> | 0.0    | 73.47 | 96.01   | 96.56 | 73.46 | 82.56   | 91.39 | 36.50    | 79.74  | 68.16 | 82.34   | 80.77 |
> | 0.01   | 73.71 | 96.11   | 96.64 | 73.66 | 82.72   | 91.60 | 36.79    | 79.83  | 68.27 | 83.48   | 80.91 |
> | 0.1    | 73.82 | 96.22   | 96.87 | 74.44 | 84.15   | 91.73 | 37.60    | 79.95  | 67.42 | 85.84   | 81.9  |
>
>
> > What if combined consistency constrain learning with other existing methods.
>
> Although we did not originally design CoPrompt as an add-on to other prompt tuning methods, the consistency constraint can be included in any method with a similar tuning mechanism. While CoCoOp constitutes a distinct prompt-tuning approach featuring a text branch conditioned on tokens generated by the image branch, the current form of the consistency constraint cannot be applied to it. However, it is feasible to integrate the consistency constraint into the CoOp method. We show the result of this experiment in the following table. Adding the constancy constraint to CoOp results in a considerable improvement in accuracy for novel classes (from 63.22 to 64.87), contributing to an overall increase in the harmonic mean (from 71.66 to 72.75).
>
> | Method   | Base Acc. | Novel Acc | HM |
> | --------- | --------- | --------- | --- |
> | CoOp      | 82.69  | 63.22 | 71.66 |
> | CoOp + Cons. | 82.80 | 64.87 | 72.75 |
>
> We have added this discussion to Appendix A.6 (page 14).
>
> > What is the training overhead in terms of time? What is training and test-time inference speed compared with prior methods?
>
> CoPrompt requires about 25\% more training time than the previous SOTA MaPLe. For example, one epoch of training on the Flower102 dataset takes 2 minutes and 9 seconds compared to 1 minute and 43 seconds for MaPLe on a single GPU. There is no overhead in the inference phase since the consistency constraint, the main component that adds the overhead, only affects the training process. We have updated the discussion on the parameters and computational complexity in Section 4.6 (Page 9) with the abovementioned information.

---

> ### Comment · Reviewer_sWY6 · 2023-11-22
>
> Dear authors,
>
> Thank you for your reply. The majority of my concerns have been addressed. Taking into consideration the feedback from other reviewers and the author's rebuttal, I decide to raise my rating.

---

> > ### Author Response · Authors · 2023-11-22
> >
> > Dear Reviewer,
> >
> > Thank you so much for your time. We are glad to find that our response addressed your concerns and that you have increased your rating.

---

### Official Review · Reviewer_D25R · 2023-10-31

**Soundness:** 3 good
**Presentation:** 3 good
**Contribution:** 2 fair
**Rating:** 5
**Confidence:** 5

**Summary:**

The paper proposes a consistency-enforced fine-tuning method for large foundation model CLIP that enables learning a new task from a few samples while maintaining the zero-shot generalizability. The proposed method incorporates the knowledge of a pretrained LLM with consistency constraints on the text branch and data augmentations on the image branch to improve the generalization further along with learnable adaptors on both image and text branches.

**Strengths:**

- The  paper is well-written and easy to follow.
- The authors have shown decent results on base-to-novel generalization.

**Weaknesses:**

- The idea of adaptors and prompt-tuning already exist in the literature. Merely combining the two ideas seems an incremental work and not novel.
- The idea of retaining the generalizability of the CLIP using consistency loss has already been explored in the paper "Self-regulating Prompts: Foundational Model Adaptation without Forgetting" (ICCV 2023) [1]. Hence,  the consistency loss doesn't contribute towards the novelty.
- The authors have not compared their approach to the above paper and there is also no reference to the paper.
- The improvements in the Domain generalization is marginal given that authors have fine-tuned the model. Same is true for cross-dataset evaluation.


[1] Muhammad Uzair Khattak, Syed Talal Wasim, Muzammal Naseer, Salman Khan, Ming-Hsuan Yang, Fahad Shahbaz Khan. Self-regulating Prompts: Foundational Model Adaptation without Forgetting. ICCV 2023 (https://arxiv.org/abs/2307.06948)

**Questions:**

- Are the vision side prompts conditioned on text side? Do authors follow MaPLe settings or Independent VL prompting?

---

> ### Author Response · Authors · 2023-11-17
>
> **Part 1**
> > The idea of adapters and prompt-tuning already exists in the literature. Merely combining the two ideas seems an incremental work and not novel.
>
> While adapters and prompt-tuning already exist in the literature (which we also acknowledged in the paper), combining them poses a considerable challenge that has not yet been solved by any prior works. This is due to the fact that the model can easily overfit the additional learnable parameters, thus losing generalizability. This phenomenon is has been reported by prior works. For example, MaPLe reported the optimal performance by adding prompts to 9 out of 12 layers of the encoder and not all 12. We highlighted the impact of the consistency constraint in our ablation study in Table 4. In the table below, we provide additional experiments emphasizing the difficulty of combining adapters with prompts for existing methods without our proposed consistency constraint.
>
> | Method/ Setup | Accuracy |
> | --------------| -------- |
> | CoPrompt      | 80.48    |
> | CoPrompt w/o Cons. const.| 78.45  |
> | MaPLe         | 78.55  |
> | MaPLe (12-layer prompt) + Adapter | 77.61 |
>
> As observed in this table, CoPrompt's performance of 80.48\% drops to 78.45\%  when prompts and adapters are trained without the proposed consistency constraint. Similarly, the optimal performance of MaPLe is 78.55\% , which drops to 77.61\%  when we increase trainable parameters in the form of prompts and adapters. This study signifies the fact that the adapters and prompts can not be combined to improve the overall accuracy without the consistency constraint proposed in our method. We have now added this discussion to Appendix A.4 (Page 14).
>
>
>
>
> > The idea of retaining the generalizability of the CLIP using consistency loss has already been explored in the paper [1]. Hence, the consistency loss doesn't contribute towards the novelty.
> > The authors have not compared their approach to the above paper and there is also no reference to the paper.
>
>
> Kindly note that we were not aware of this work at the time of submission since this paper was accepted in ICCV 2023, which was around the same time that we were preparing for our submission to ICLR. It is worth noting that our paper had been publicly available on arXiv before PromptSRC. Our paper was posted on arXiv on Thu, 1 Jun 2023, while PromptSRC was posted on Thu, 13 Jul 2023. However, while both PromptSRC and our proposed CoPrompt share a common goal of ensuring consistency, the proposed solutions in the two methods differ significantly. These distinctions are as follows:
>
>
> - While prompts are the only training parameters in PromptSRC, we introduced the concept of integrating adapters with prompts. It provides additional tunable parameters for adopting CLIP to a new domain, and training under the influence of consistency helps the model to improve generalization without over-fitting the few labelled samples.
>
> - On the language branch, PromptSRC uses a pool of text templates to increase the number of text prompts, which are all hand-designed prompts. In contrast, our proposed solution uses an LLM to generate a more descriptive sentence. From the ablation study on PromptSRC and CoPrompt, we see a larger improvement with our approach.
>
> - For the consistency constraint, PromptSRC used L1 loss, whereas our CoPrompt utilizes a cosine loss. As opposed to L1 loss, cosine loss captures the angular similarity between vectors rather than relying solely on their magnitudes. Nonetheless, we already explored L1, and also MSE as the consistency loss, which was presented in Table 5(b). The results showed comparatively better performance for cosine loss over L1 or MSE.
>
> - PromptSRC utilized independent prompts on language and vision branches, while CoPrompt employs a multi-modal prompt with a coupling function between language and vision branches.
>
> - We now directly compare our results to PromptSRC in the following tables. On the base-to-novel generalization task, our method performs better than PromptSRC in the evaluation of 11 datasets, with an average HM of 80.48 vs.  PromptSRC's 79.97. Similarly, on the cross-dataset evaluation, CoPrompt outperforms PromptSRC by 1.19\%.

---

> ### Author Response · Authors · 2023-11-17
>
> **Part 2**
>
>
> | Method | Base Acc. | Novel Acc | HM |
> | ------ | --------- | --------- | --- |
> | CoPrmpt (ours)  | 84.00 | 77.23 | 80.48 |
> | PromptSRC | 84.26 | 76.10 | 79.97 |
>
>
>
> | Method          | Caltech | Pets  | Cars  | Flowers | Food  | Aircraft | SUN397 | DTD   | EuroSAT | UCF   | Average |
> |-----------------|---------|-------|-------|---------|-------|----------|--------|-------|---------|-------|---------|
> | CoPrompt        | 94.50   | 90.73 | 65.67 | 72.30   | 86.43 | 24.00    | 67.57  | 47.07 | 51.90   | 69.73 | 67.00   |
> | PromptSRC | 71.27 | 93.60 | 90.25 | 65.70 | 70.25 | 86.15 | 23.90 | 67.10 | 46.87 | 45.50 | 68.75 | 65.81 |
>
> We now discuss the similarity of our method with PromptSRC in the Method section (Page 4, Paragraph 3), and also expand the results in Tables 1, 2 and 3 (Pages 6 and 7) to compare with PromptSRC.
>
>
> > The improvements in the Domain generalization is marginal given that authors have fine-tuned the model. Same is true for cross-dataset evaluation.
>
> While the average improvement in cross-dataset evaluation and domain adaptation is relatively lower compared to the improvements in few-shot and zero-shot performance, certain datasets exhibit notable progress. For example, EuroSAT demonstrated a notable improvement of 3.84\% in cross-dataset evaluation. Kindly note that the improvements of existing methods in these two evaluations were on a comparable scale. To illustrate, MaPLe demonstrated enhancements of 0.56\% and 0.15\% for cross-dataset evaluation and domain generalization, respectively. Similarly, PromptSRC (accepted in ICCV'23) exhibited a 0.38\% improvement in domain generalization, but a 0.49\% drop in cross-dataset evaluation.
>
>
> > Are the vision side prompts conditioned on text side? Do authors follow MaPLe settings or Independent VL prompting?
>
> For the implementation of the prompts, we followed the same approach as used in MaPLe.

---

> ### Comment · Reviewer_D25R · 2023-11-21
>
> Dear authors, thank you for addressing my comments
>
> - I am satisfied with the explanation of using prompts and adapters jointly. Authors have done a nice job.
> - Also, the differences between PromptSRC and author's work seem acceptable.
>
> However, I still  feel the work lacks content in terms of novelty. Based on the responses, I have increased my rating accordingly.
>
> Thank you!

---

> > ### Author Response · Authors · 2023-11-22
> >
> > Dear Reviewer,
> >
> > Thank you for your time and for your attention to our rebuttal. We’re pleased that our rebuttal has addressed your comments and led to an increased rating. Our work not only improved overall performance but also introduced several concepts that had not been explored before. Specifically, our proposed consistency constraint is a novel concept which allows for adopting the pre-trained model to a new domain without losing its generalization. Also, the concept of training adapters and prompts together had not been successfully explored due to the over-fitting issue (as discussed in our previous rebuttal response), which allows us to improve performance on the downstream task. All these resulted in a significant improvement in base-to-novel generalization and cross-dataset evaluation. Our detailed analysis shows the effectiveness of each of these components.
> >
> > In the revised manuscript, we also show that the proposed consistency constraint can be used as an add-on to existing methods like CoOp and improve its performance. We have added additional experiments to show that CoPormpt also shows improvement for more recent methods like EVA-CLIP. We hope this discussion further highlights the novelty and our contribution in this work.

---

### Official Review · Reviewer_iShv · 2023-11-01

**Soundness:** 3 good
**Presentation:** 3 good
**Contribution:** 3 good
**Rating:** 6
**Confidence:** 5

**Summary:**

This paper proposes a new adaptation method for CLIP like large scale vision-language models for generalization benchmarks. Specifically, the authors propose 3 techniques to improve generalization of CLIP. Firstly, they observe that the main cause of poor generalization is the lack of consistency constraints between the learned embeddings and the original pretrained embeddings. To overcome this issue, consistency losses are used at the text side as well as the image side separately. Secondly, the inputs to the original models are perturbed with the help of augmentations and LLM captions for image and text side respectively. Lastly, the proposed method combines the adapter and prompt learning modules with-in the same architecture for improved performance.

Extensive benchmark comparisons are conduced on 3 different generalization tasks where the proposed approach shows improvements against prior methods. Furthermore, ablation studies are provided for analyzing contributions of each component separately and motivating the design choices.

**Strengths:**

1) This paper addresses an important aspect of generalization of pre-trained CLIP like models for downstream task adaptation. Most of the prior methods struggles to achieve good performance on unseen classes and datasets, while this method explicitly add training constraints to mitigate the issue.
2) The proposed framework is motivated fairly, and the strength of its individual components have been demonstrated clearly in the ablation studies.
3) The method shows impressive performance against the previous prompt learning methods.
4) Paper is easy to read.

**Weaknesses:**

1. The authors mentioned that their baseline is MaPLe, which uses coupling functions between vision and text branches, but in Figure 3, no coupling functions are visible. It will be good to clarify the exact architecture used in the proposed framework. Also I think there is graphic error in image encoder as the visual prompts (orange color) are not shown in intermediate layers of CLIP visual encoder.

2. It will be good to see the proposed method generalization for a newer V-L model. CLIP is relatively outdated and the authors are encouraged to show result on at least another recent CLIP variant. For example on EVA-CLIP[1] model.

3. There is a recent prompt learning method PromptSRC [2], which also seems to introduce consistency constraints to prompt learning to improve generalization. How is the proposed method different from this work? Also, all fair comparisons should be added in the main paper.

4. The diagrams in the paper are of very poor quality. Specially the text in the Figure 2. graph is very small and the color scheme used is confusing. Also in the Figure 1, their is no indication of using adapters in Fig. 1b.

5. I think there is some writing logical errors in the paper. For example, in the Adapters heading in section 3, adapter based method are being mentioned but prompts have been written instead of the adapter blocks.

[1] Exploring the Limits of Masked Visual Representation Learning at Scale (CVPR-23)
[2] Self-regulating Prompts: Foundational Model Adaptation without Forgetting (ICCV-23)

**Questions:**

Please refer to the weaknesses section for additional questions and queries!

---

> ### Author Response · Authors · 2023-11-17
>
> **Part 1**
>
> > In Figure 3, no coupling functions are visible. Visual prompts (orange color) are not shown in intermediate layers of CLIP visual encoder.
>
> Thank you for pointing this out. We have now visualized the coupling function and fixed the colour in the visual branch, in Figure 2 (Page 5).
>
> > It will be good to see the proposed method generalization for a newer V-L model. For example, on EVA-CLIP [1] model.
>
> In the original submission, we only focused on CLIP since all existing methods we compared had reported their results on the same encoder. As per your suggestion, we have now investigated the performance of the proposed method, as well as the previous SOTA MaPLe, with the EVA-CLIP backbone. The results are presented below:
>
> | Method   | Backbone      | Base Acc  | Novel Acc | HM    |
> |----------|---------------|-----------|-----------|-------|
> | MaPLe    | CLIP-B/16     | 82.28     | 75.14     | 78.55 |
> | CoPrompt | CLIP-B/16     | 84.00     | 77.23     | 80.48 |
> | MaPLe    | EVA-CLIP-B/16 | 82.65     | 75.78     | 79.06 |
> | CoPrompt | EVA-CLIP-B/16 | 84.29     | 77.50     | 80.75 |
>
> This table demonstrates that CoPrompt also works well on the EVA-CLIP backbone and shows similar improvements over MaPLe. Overall, the EVA-CLIP encoder performs slightly better than the CLIP encoder. We have added this discussion to Appendix A.3 (page 14).
>
>
>
> > There is a recent prompt learning method PromptSRC [2]. How is the proposed method different from this work?
>
> Kindly note that we were not aware of this work at the time of submission since this paper was accepted in ICCV 2023, which was around the same time that we were preparing for our submission to ICLR. It is worth noting that our paper had been publicly available on arXiv before PromptSRC. Our paper was posted on arXiv on Thu, 1 Jun 2023, while PromptSRC was posted on Thu, 13 Jul 2023. However, while both PromptSRC and our proposed CoPrompt share a common goal of ensuring consistency, the proposed solutions in the two methods differ significantly. These distinctions are as follows:
>
>
> - While prompts are the only training parameters in PromptSRC, we introduced the concept of integrating adapters with prompts. It provides additional tunable parameters for adopting CLIP to a new domain, and training under the influence of consistency helps the model to improve generalization without over-fitting the few labelled samples.
>
> - On the language branch, PromptSRC uses a pool of text templates to increase the number of text prompts, which are all hand-designed prompts. In contrast, our proposed solution uses an LLM to generate a more descriptive sentence. From the ablation study on PromptSRC and CoPrompt, we see a larger improvement with our approach.
>
> - For the consistency constraint, PromptSRC used L1 loss, whereas our CoPrompt utilizes a cosine loss. As opposed to L1 loss, cosine loss captures the angular similarity between vectors rather than relying solely on their magnitudes. Nonetheless, we already explored L1, and also MSE as the consistency loss, which was presented in Table 5(b). The results showed comparatively better performance for cosine loss over L1 or MSE.
>
> - PromptSRC utilized independent prompts on language and vision branches, while CoPrompt employs a multi-modal prompt with a coupling function between language and vision branches.
>
> - We now directly compare our results to PromptSRC in the following tables. On the base-to-novel generalization task, our method performs better than PromptSRC in the evaluation of 11 datasets, with an average HM of 80.48 vs.  PromptSRC's 79.97. Similarly, on the cross-dataset evaluation, CoPrompt outperforms PromptSRC by 1.19\%.
>
>
> | Method | Base Acc. | Novel Acc | HM |
> | ------ | --------- | --------- | --- |
> | CoPrmpt (ours)  | 84.00 | 77.23 | 80.48 |
> | PromptSRC | 84.26 | 76.10 | 79.97 |
>
>
>
> | Method          | Caltech | Pets  | Cars  | Flowers | Food  | Aircraft | SUN397 | DTD   | EuroSAT | UCF   | Average |
> |-----------------|---------|-------|-------|---------|-------|----------|--------|-------|---------|-------|---------|
> | CoPrompt        | 94.50   | 90.73 | 65.67 | 72.30   | 86.43 | 24.00    | 67.57  | 47.07 | 51.90   | 69.73 | 67.00   |
> | PromptSRC | 71.27 | 93.60 | 90.25 | 65.70 | 70.25 | 86.15 | 23.90 | 67.10 | 46.87 | 45.50 | 68.75 | 65.81 |
>
> We now discuss the similarity of our method with PromptSRC in the Method section (Page 4, Paragraph 3), and also expand the results in Tables 1, 2 and 3 (Pages 6 and 7) to compare with PromptSRC.

---

> ### Author Response · Authors · 2023-11-17
>
> **Part 2**
>
> > The diagrams in the paper are of very poor quality.
>
> As per your suggestion, we improved the figures to make them visually more appealing. Also, we have now visualized the adapters in Figure 1 (Page 1).
> However, we removed Figure 2 due to its overall quality and redundancy in presenting information already available in Table 1, and to make room for the revisions based on the reviews.
> We would appreciate it if the reviewer could kindly point out any further changes that they deem necessary, and we will happily make those additional changes.
>
>
>
> > I think there are some writing logical errors in the paper. For example, in the Adapters heading in section 3, adapter based method are being mentioned but prompts have been written instead of the adapter blocks.
>
> We checked the writing of the mentioned section, and are glad to inform you that there are no writing errors. However, we have revised the writing to resolve this confusion. The sentence mentioning 'prompt' referred to an existing work (Khattak et al., 2022) that discussed the fact the additional parameters (whether prompt or adapter) introduces the risk of overfitting. We have now revised the description on Page 5, Paragraph 1 to say: 'For instance, Maple achieved the best performance when adding learnable parameters to only nine out of twelve layers of the CLIP backbone.'

---

> > ### Comment · Reviewer_iShv · 2023-11-18
> >
> > Dear authors, thank you for providing a good rebuttal.
> >
> > The manuscript with the updated version looks more comprehensive. CoPrompt shows improvements with recent CLIP models like EVA-CLIP, which is a good sign.
> >
> > Although comparing PromptSRC release date on arXiv with CoPrompt does not show that CoPrompt was the earlier work (because it looks like PromptSRC was released right after ICCV results, so the work must have been done long before), I think it is fine and both works contribute towards solving important challenges in the vision community.
> >
> > Overall, I am satisfied with the revised version of the paper and will keep my current score of above acceptance threshold.
> >
> > Thank you!

---

> > > ### Author Response · Authors · 2023-11-21
> > >
> > > Dear Reviewer,
> > >
> > > Thank you so much for reading the rebuttal and responding to it.
> > > We are glad to find that the rebuttal response was satisfactory to you and that you are supporting the acceptance of our paper.

---

### Author Response · Authors · 2023-11-17

We sincerely thank the review committee for their time and for providing constructive feedback. We are happy to see the overall engaging comments given by all the reviewers. We have carefully addressed all the concerns raised by the reviewers under the individual response section. We have also **updated the main PDF**. Following, we provide a *summary* of our responses.

- **Effectiveness on other V-L backbones:** As per the suggestion of **Reviewer iShv**, we include additional experiments to show that the proposed solution generalizes to other V-L backbones, such as EVA-CLIP in **Table 10**.

- **Tuning adapters and prompts together:** To address the concern of **Reviewer D25R**, we discussed the difficulties that arise when we increase trainable parameters by combining adapters with prompts. While it appears to be promising and straightforward to train both adapters and prompts together, we show with additional experiments (please see **Table 11**) that training these additional parameters without our proposed consistency constant yields lower performance.

- **Relation to PromptSRC:** This paper was pointed out by **Reviewers iShv and D25R**. Kindly note that our paper had been publicly available on arXiv before PromptSRC, i.e., our paper was posted on arXiv on Thu, 1 Jun 2023, while PromptSRC was posted on Thu, 13 Jul 2023. Nonetheless, we have now pointed out four key differences between PromptSRC and CoPrompt. We now also directly compare our results in **Tables 1, 2, and 3**, where we observe that CoPrompt performs better than PromptSRC in the base-to-novel generalization and cross-dataset evaluations.

- **Comparison to zero-shot performance of CLIP:**
In response to the suggestion from **Reviewer sWY6**, we have included additional experiments to ensure a fair comparison with the zero-shot performance of CLIP. To this end, we show the results for the following two scenarios: (1) removing the LLM-generated prompts from CoPrompt, which performs better than zero-shot performance of CLIP by 8.99\% (please see **Table 5(c)**); and (2) adding LLM-generated prompts to CLIP for its zero-shot evaluation (please see **Table 12**), which drops CLIP's zero-shot performance by 2.56\%.

- **Impact of lambda:** As per the suggestion of **Reviewer sWY6**, we now conduct a sensitivity study on lambda and present the results in **Table 6**.

- **Impact of the proposed consistency constraint on other tuning methods:** As per the suggestion of **Reviewer sWY6**, we now present further experiments to show that the proposed consistency constraint can be used as an add-on to other tuning methods to improve their performance. We present the results in **Table 13**. This experiment shows that the performance of CoOp is considerably improved when consistency constraint is applied.

- **Improving the figures:** As per the suggestion of **Reviewer iShv**, we have revised Figures 1 and 2 for more clarity and to better demonstrate our proposed method.

---

### Meta-Review · Area_Chair_nKMk · 2023-12-06

**Metareview:**

The paper introduces a consistency-guided prompt learning method (CoPrompt) for improving the generalization of CLIP-like models, addressing an important aspect of model adaptation. Despite initial concerns about its similarity to PromptSRC, the authors have effectively clarified the differences and addressed most reviewer concerns during the rebuttal stage. The decision to accept the paper as a poster is based on its demonstrated improvements in generalization tasks, well-motivated framework, and effective ablation studies, despite some incremental aspects in methodology.

**Justification For Why Not Higher Score:**

The incremental nature of combining existing ideas like adaptors and prompt-tuning, and concerns about the novelty of consistency loss, which has been explored in prior work, limit the score.

**Justification For Why Not Lower Score:**

The paper's strengths lie in its clear presentation, significant numerical results, and effective use of consistency constraints to mitigate overfitting. The method's performance in base-to-novel generalization and cross-domain settings, as well as the authors' successful rebuttal addressing the reviewers' initial concerns, justify a score that supports acceptance.

---

### Decision · Program_Chairs · 2024-01-16

Accept (poster)